# Role of Anti-Cancer Peptides as Immunomodulatory Agents: Potential and Design Strategy

**DOI:** 10.3390/pharmaceutics14122686

**Published:** 2022-12-01

**Authors:** Amit Kumar Tripathi, Jamboor K. Vishwanatha

**Affiliations:** Department of Microbiology, Immunology and Genetics, University of North Texas Health Science Center, Fort Worth, TX 76107, USA

**Keywords:** ACPs (anti-cancer peptides), peptide drugs, immunomodulation

## Abstract

The usage of peptide-based drugs to combat cancer is gaining significance in the pharmaceutical industry. The collateral damage caused to normal cells due to the use of chemotherapy, radiotherapy, etc. has given an impetus to the search for alternative methods of cancer treatment. For a long time, antimicrobial peptides (AMPs) have been shown to display anticancer activity. However, the immunomodulatory activity of anti-cancer peptides has not been researched very extensively. The interconnection of cancer and immune responses is well-known. Hence, a search and design of molecules that can show anti-cancer and immunomodulatory activity can be lead molecules in this field. A large number of anti-cancer peptides show good immunomodulatory activity by inhibiting the pro-inflammatory responses that assist cancer progression. Here, we thoroughly review both the naturally occurring and synthetic anti-cancer peptides that are reported to possess both anti-cancer and immunomodulatory activity. We also assess the structural and biophysical parameters that can be utilized to improve the activity. Both activities are mostly reported by different groups, however, we discuss them together to highlight their interconnection, which can be used in the future to design peptide drugs in the field of cancer therapeutics.

## 1. Introduction

Rudolf Virchow first observed that persistent inflammation transcends into cancer and tumor tissues show a large infiltration of inflammatory cells. Later on, Dvorak showed that carcinogenesis and inflammation have commonalities in terms of proliferation, migration, cytokine secretion and angiogenesis. He described cancer as a “wound that does not heal” [1]. In modern times, the three main methods of cancer treatment so far have been Chemotherapy, radiotherapy and immunotherapy. Chemotherapy is the most established method of treatment that kill fast-dividing cancer cells. However, most cancer drugs have very poor cell selectivity and kill normal cells along with cancer cells indiscriminately [2,3]. Moreover, continuous use of this therapy increases the possibility of drug resistance in the body along with the chances of recurrence. Radiotherapy is the second type of cancer therapy that uses high-energy beams to kill cancer cells. X-rays are the most commonly used energy beams but, protons or other types of energy can also be used. Unfortunately, radiotherapy causes collateral damage as, despite the advancements in modern types of equipment, the radiation kills normal cells along with the targeted cancer cells. Immunotherapy is the third kind of therapy that improves the patient’s immune system to exert an anti-tumor effect. This treatment method has fewer side effects than chemotherapy and radiotherapy, and the therapeutic effects are long-lasting [4]. As a standard job, the immune system detects cancerous cells due to the presence of abnormal cell surface markers. Biopsies of patients show various immune cells in and around tumors [5]. These cells, called tumor-infiltrating lymphocytes or TILs, are a sign that the immune system is responding to the tumor. Cancer patients whose tumors contain TILs often have a lesser level of cancer severity than those whose tumors do not contain them. The interrelationship between inflammation, innate immunity and cancer is well known [6]. Persistent inflammation triggers cancer initiation that is characterized by infiltration of mononuclear immune cells (including macrophages, lymphocytes, and plasma cells), tissue destruction, fibrosis, and increased angiogenesis [7].

## 2. Where Anti-Cancer Peptides Stand

The discovery of peptide hormone insulin gave an impetus to peptide therapeutics. During the last two decades, peptides have grown as encouraging healing mediators in many areas such as diabetes [8], cardiovascular diseases [9] and cancer treatment [10] (Figure 1). Improvements in peptide design have increased its applications in other fields as well [11,12]. The beginning of the 21st century witnessed rapid advancements in various interdisciplinary fields like analytics, structural biology Computer-assisted drug discovery, and bioinformatics tools which have made peptide design much easier and also minimized the chances of drug failure. This has led to the opening of new areas of drug discovery where peptide synthesis, chemical modifications and the evaluation of biological activities can be done simultaneously that speed up the process of lead molecule identifications. Insights in the global market for peptide therapeutics is projected to record a value of USD 44.43 billion in 2026, progressing at a CAGR of 6.95%, over the period 2022–2026. Currently, more than 80 peptide-based drugs are present in the market for the treatment of a wide range of diseases including cancer, osteoporosis, diabetes, etc. [13]. It is estimated that up to 400–600 peptide drugs are in preclinical trials. After 2017, USFDA has already approved more than 10 peptide-based drugs. Of these, LupkynisTM and Zegalogue were recently approved in 2021, while ImcivreeTM, Victoza, LUPRON DEPOT, Zoladex, Sandostatin and Somatuline received approval in 2020 [14]. The three drugs which have touched global sales of over $1 billion are goserelin, leuprolide and octreotide [15]. Goserelin is a synthetic decapeptide analog of luteinizing hormone-releasing hormone (LHRH). It has anti-cancer activity and is used in treatments of both breast and prostate cancer [16] Leuprolide is a peptide analog of gonadotropin-releasing hormone (GnRH) which is used as a palliative treatment of prostate cancer and many other conditions [17]. Studies have shown that it also possesses an immunomodulatory effect. The investigations showed that Leuprolide Acetate (LA) administration to Experimental autoimmune encephalomyelitis (EAE) rats considerably inhibited the activation of NF-κB which is central to both inflammation and cancer progression. It was shown that treatment with LA reduced the production of TNF-α, IL-1β and other inflammatory cytokines which play an important role in both inflammation and cancer initiation [18]. The third successful peptide drug Octreotide is an analog of somatostatin. It helps to temporarily reduce the tumor size and diminish cancer development. Due to its large therapeutic abilities, octreotide has evolved as a backbone of clinical cancer therapeutics [19,20]. With the advent of bioinformatics tools, various short peptides have been identified from the naturally occurring proteins that have shown a high affinity for cancer cells. The structure-function relationships of anti-cancer peptides showed that certain biophysical parameters are present in these peptides that attract them to the cancer cells. Based on these parameters, several naturally occurring and synthetic peptides are being identified that show anti-cancerous properties.

## 3. Naturally Occurring AMPs/ACPs with Immunomodulatory Activities

The helicity is strongly correlated with antimicrobial and anti-cancer activity [21]. Naturally occurring anti-microbial peptides have shown potent anti-cancer activities in in vitro experiments in several independent studies (Table 1). Besides, naturally occurring proteins that are associated with cancer progression pathways have been used to design immunomodulatory peptides that can inhibit LPS-mediated inflammatory responses [22,23]. LL-37, Magainin-II, Melittin and other naturally occurring anti-microbial peptides have shown appreciable anti-cancer activity [24,25,26]. Additionally, they contain good anti-inflammatory activity which is an important immunomodulatory activity to tackle the dual problem of cancer and inflammation [27]. Melittin (GIGAVLKVLTTGLPALISWIKRKRQQ), a peptide first isolated from bee venom has shown to be a promising candidate for cancer therapy. It is an alpha-helical anticancer peptide that plays a key role in immunomodulation and inhibits proinflammatory agents. The anti-cancer properties exerted by melittin are very similar to anti-cancer drugs and involve cell cycle arrest, anti-proliferative activity on cancer cells and activation of caspases [28]. Besides these, melittin is also shown to induce apoptosis in cancer cells through ROS generation and the diffusion of mitochondrial membrane potential [29].

Unfortunately, melittin is a non-cell selective peptide and displays cytotoxicity to normal cells as well [30]. Therefore, to achieve its true therapeutic potential, suitable analogs of it are to be designed that show reduced cytotoxicity to normal cells but retain the anti-cancer activity. Asthana et al. have identified a leucine zipper motif in melittin which can be used as a switch to design bioactive analogs [31]. These analogs can be used for their anti-cancer and immunomodulatory activities but with reduced cytotoxicity to normal cells. Srivastava et al. showed that melittin can neutralize the lipopolysaccharide-induced proinflammatory pathways in RAW 264.7 and primary macrophage cells and the leucine zipper motif present in the peptide played an important role in its immunomodulatory activity [32]. Liu et al. created a bifunctional fusion protein melittin-MIL-2, which was a recombinant of melittin and a mutant IL-2 [33]. The melittin-MIL-2 displayed potent anti-cancer activities in comparison to Melittin and rIL-2 alone. The MIL-2 displayed anti-proliferative activity against cancer cells derived from different tissues. In the in vivo experiments, the MIL-2 was able to inhibit the tumor growth in liver, lung and ovary cancer cells. The investigators also showed that exposure of MIL-2 was able to reduce the ability of breast cancer cells to metastasize to the lungs. Another promising antimicrobial peptide that can be used for the dual role of anticancer and immunomodulatory activities is Magainin II (KWKLFKKIKFLHSAKKF). It was first isolated from the skin of *Xenopus laevis* frogs. Studies have shown that magainin II inhibited the cell proliferation of bladder cancer cells while did not cause any toxicity to normal fibroblast 3T3 cells [34]. Although Magainin II did not show effective anti-cancer activity on human breast cancer cells MDA-MB-231, it displays good anti-cancer activity against lung cancer cell line A549 [25]. It is not toxic to human immortalized epidermal cells under similar conditions. While Magainin II itself does not cause any immunomodulatory activity, hybrid peptides designed using cecropin A17 and Magainin II showed potent anti-cancer and anti-inflammatory activity [35,36]. A cecropin A–magainin II hybrid peptide called P18 (KWKLFKKIPKFLHLAKKF) was effective against human leukemia K562 cells [37]. Tang et al. showed that P18 induced necrosis in these cells instead of activating the apoptotic pathway. The mechanism of action of the peptide involved the diffusing of the plasma membrane potential in the cells after peptide exposure [37]. Nan et al. while studying the immunomodulatory activity of P18 showed that when mouse macrophage cell line RAW264.7 was challenged with LPS from *E. coli* in the absence or presence of P18, it inhibited the LPS-mediated production of pro-inflammatory mediators and cytokines viz. nitrite (NO), TNF-α and IL-1β [38]. This was a classic example of using two naturally occurring peptide sequences to design a hybrid peptide with immunomodulatory activity, which was not present in the parent peptide. The judicious substitution of key residues at either terminus of an anti-cancer peptide often improves its biological activity [39]. However, it is essential to make a conservative replacement so that the biophysical parameters like charge, hydrophobicity, etc. remain similar to the parent peptide. Along similar lines, Arias et al. designed improved analogs of tritrpticin (VRRFPWWWPFLRR) for potent anti-cancer activity against the Jurkat leukemia cell line [40]. By designing a series of analogs of tritrpticin, they found that if the arginines at both the termini are replaced with lysines or lysine-derivatives, it improves the cell-selectivity of the peptides towards Jurkat leukemia cells as opposed to normal peripheral blood mononuclear cells (PBMCs). Interestingly, arginine to lysine substitution also enhanced the biological activity in other sequentially similar peptides including indolicidin (ILPWKWPWWPWRR) and puroindoline A (PuroA FPVTWKWWKWWKG-NH2) [41]. Ghiselli et al. checked the anti-inflammatory activity of Indolicidin, a closely related peptide to tritrpticin in two rat models of polymicrobial peritonitis. The investigators used two different models to induce sepsis. One by intraperitoneal injection of LPS and the other by using cecal ligation and puncture (CLP model) of inflammation. The results showed that indolicidin treated group decreased the bacterial burden in visceral organs like peritoneum, spleen and liver and plasma levels of LPS-mediated production of TNF-α and IL-6 were also inhibited. This signifies the dual role of the peptide as both an immunomodulatory and anticancer peptide in addition to being a well-established antimicrobial peptide [42].

Anti-cancer peptides with immunomodulatory activities have also been reported from marine sources [43]. Marine fishes are rich sources of anti-cancer peptides possessing immunomodulatory activities [44]. Marine animals possess poor immune systems and live in ecological niches where they are exposed to diverse pathogens. Iijima et al. purified Chrysophsins in marine fish *Chrysophsis major*. There are three different forms of chrysophsins [45]. The majority of the anti-cancer and immunomodulatory studies are done on two isoforms, Chrysophsin-I (FFGWLIKGAIHAGKAIHGLIHRRRH) and Chrysophsin-II (FFGWLIRGAIHAGKAIHGLIHRRRH). Hsu et al. checked the anti-cancer activity of Chrysophsin-1. Their results showed that the peptide followed a lytic mechanism to kill cancer cells mostly through pore formation. The inhibition ratio was less for normal cell lines viz. NIH-3T3 and WS-1 [46]. Tripathi et al. identified the GXXXXG motif in Chrysophsin I and designed various proline-substituted analogs of the Chrysophsin-1 and showed that the peptides in addition to the anti-cancer activity as described by other workers also possess potent immunomodulatory activity. The authors were able to show that one of the proline-substituted analogs rescued the mice from the lethal dose of LPS [47]. Temporin L (FVQWFSKFLGRIL), another highly studied alpha-helical anti-microbial peptide has significant anti-cancer and immunomodulatory activities [48]. Swithenbank et al., while studying the activity of Temporins and Bombinin H2 (LIGPVLGLVGSALGGLLKKI) on lung cancer cell lines reported that Temporin L exposure to cancer cell lines viz. A-549 and Calu-3 caused significant cytotoxicity in a dose-dependent manner [49]. Srivastava et al. investigated the immunomodulatory activity of Temporin L. They identified a phenylalanine zipper in Temporin L that can be used to design Temporin L analogs that are less toxic to normal cells and exhibit anti-endotoxin activities [50]. Studies have shown that Temporin L directly binds to LPS and can be a therapeutic agent in septic shock [51].

Interestingly, phenylalanine heptad repeats have also been used to design synthetic peptides that contain potent anti-cancer and immunomodulatory activities. Tripathi et al. showed that if the phenylalanine residues are replaced with proline in a synthetic peptide designed on the basis of phenylalanine heptad repeats, the resultant peptide exhibit potent anti-cancer and immunomodulatory activities. The proline substituted analogs of parent peptide designed on phenylalanine heptad repeats also inhibit migration in MDA-MB-231 breast cancer cells and induce programmed cell death by activating the intrinsic pathway of apoptosis. The same peptides contained anti-endotoxin activities as they inhibit the LPS-mediated NF-kB nuclear translocation and inhibit the production of pro-inflammatory cytokines [52]. Hepcidin (ICIFCCGCCHRSKCGMCCKT) is also a good example of an anti-cancer peptide containing immunomodulatory activity [53]. Cytotoxicity data of hepcidin on myeloma cells indicate that it causes plasma membrane damage and DNA fragmentation in these cancer cells to exhibit its anti-cancerous activities [54]. An independent study about the immunomodulatory activity of hepcidin showed that it can up-regulate the expression of both pro- and anti-inflammatory cytokines like TNF-α, IL-1β, and IL-10 in teleost leukocytes. [53]. The mRNA expression was also found to be high in the organs like the spleen and head kidney. LL-37(LLGDFFRKSKEKIGKEFKRIVQRIKDFLRNLVPRTES) is probably the most widely studied Cathelicidin. Originally investigated for its antimicrobial activities, LL-37 soon was reported to inhibit a wide range of cancers and is very context-specific [55]. The role of LL-37 in colon cancer seems to be most interesting. A differential expression of LL-37 has been observed in normal colon and cancer colon mucosa. It had been reported that LL-37 gets downregulated as colon cancer progresses [56]. This has led to the idea of using LL-37 as a colon cancer biomarker [57]. It can also induce apoptosis by upregulating the levels of Bax/Bak and downregulating the BCL-2 levels [58]. LL-37 has also been shown to increase the PUMA (p53 upregulated modulator of apoptosis) expression which is a modulator of apoptosis in colon cancer cells [24]. Besides this, LL-37 also increases the nuclear translocation of apoptosis-inducing-factor (AIF) and endonuclease G (EndoG) in colon cancer cells to induce apoptosis [56]. FK-16, a derivate of full-length LL-37 containing the same amino acid residues from 17 to 32 followed a similar mechanism as the parental LL-37 to cause apoptosis by increasing the nuclear levels of apoptosis-inducing-factor (AIF) and endonuclease G (EndoG) in colon cancer cells in a caspase-independent manner [59]. LL-37 also has a very potent immunomodulatory activity on different types of immune cells. It is shown to have antisepsis properties and has been proven to neutralize the inflammatory responses activated by bacterial components like LPS and LTA. Culturing the bone marrow-derived macrophages with LPS with or without LL-37 showed that the LL-37 was able to almost completely cancel out the LPS-mediated TNF-α and brought it to an almost basal level [60].

### 3.1. Amino Acid Arrangement and Their Biophysical Parameters Determine Anti-Cancer and Immunomodulatory Properties

Amino acid residues in an anti-cancer peptide can dictate its cell permeability. Leucine, lysine, histidine and glycine are abundant in peptides having anti-cancer and immunomodulatory activity [61]. Studies have reported that electrostatic interaction between cationic amino acid residues and negatively charged components of cancer cell membrane mostly phosphatidylserine is the first interaction to begin the anti-cancer activity [62]. Similarly, anti-cancer peptides eg. Piscidin-1 exhibiting anti-endotoxin activity interact with lipopolysaccharide (LPS) on similar lines [63,64]. The LPS molecules also contain anionic groups making them ideal candidates for electrostatic interactions by cationic ACPs [65]. Glutamic and aspartic acid are shown to inhibit cell proliferation in hepatoma cells and inhibit the AKT phosphorylation, a key signal transduction protein in cancer biology [66]. Proline in anti-cancer peptides impart proteolytic stability, aids in membrane interaction and brings conformational changes to the secondary structure of the peptides [67]. Phenylalanine imparts hydrophobicity to the peptides, which is an important biophysical parameter for both anti-cancer and immunomodulatory activity. It has been observed that the incorporation of phenylalanine amino acid sometimes improves the anti-cancer activity of the peptides like Galaxamide [68]. Tyrosine and tryptophan are hydrophobic amino acids, which is an important biophysical attribute for anti-cancer activity. Tryptophan plays an important role in the anti-cancer activity of peptides such as indolicidin and trans-activator of transportation (TAT)-Ras GTPase-activating protein-326 peptides [61,69]. Tryptophan contributes to the cell-penetrating ability of the ACP to facilitate its entry to the cell involving an endocytic pathway and DNA-binding [70,71]. Thus, it has been shown that cationic and hydrophobic amino acid residues are critical to both anti-cancer and immunomodulatory activity of the peptides and play an important role in the preliminary interaction of the peptide and cell membrane interactions [72,73,74,75,76].

### 3.2. Knowledge of Structural Determinants of AMPs/ACPs Are Same as for Immunomodulatory Peptides

There are several bio-physical parameters in AMPs, which are usually present in anti-cancer peptides, and their proper knowledge can be used to design ACPs without using any computer or software applications. By judiciously choosing the amino acids at specific positions and utilizing the structural determinants like charge, hydrophobicity, etc. potent antimicrobial peptides could be designed which possess the important attribute of cell selectivity.

#### 3.2.1. Size

ACPs despite similar biological behaviors can vary in terms of length. The number of residues in the amino acid sequence of ACPs ranges from 5 to lesser than 100, however the most ACPs fall in the range of 15–50 residues [77]. They can be as small as KTH-222, having a length of 8 amino acids (NH_2_-LKGQLRCI-C0_2_H), or as long as LL-37 and PR-39 [78,79,80].

#### 3.2.2. Amino Acid Prevalence

Biological behavior of an ACP is a mere manifestation of its structural components, and these parameters depend on the amino acid sequence and prevalence of specific amino acids in the peptide sequence [81]. It is observed that many anti-cancer and immunomodulatory peptides more often contain basic residues like lysine or arginine than acidic residues like aspartate and glutamate [82,83]. On the other side, hydrophobic residues such as alanine, leucine, isoleucine, phenylalanine and tryptophan are well represented in ACPs and contribute to acquiring a stable conformation in a membrane environment [84]. L-K6 and K4R2Nal2-S1 are examples of such peptides [85,86].

#### 3.2.3. Charge

The presence of positive charge is one of the most important parameters to initiate electrostatic interactions with negatively charged cancer cell membranes, and encourage self-promoted uptakes of ACPs [86]. Hence, it is not surprising that most of the cationic ACPs target the anionic membranes of cancer cells. Apart from many natural anti-cancer peptides, synthetic peptides like IK-13 and LK-13 reported by Hadianamrei et al. were designed chiefly based on positive charge [87].

#### 3.2.4. Conformation

Based on the amino acid composition and their positions, ACPs acquire different secondary structures including α- helices, β-sheet, loops and extended helical conformations [87]. Amphipathic α-helical peptides are the most prevalent, followed by β-sheet peptides [88,89,90]. However, a large majority of ACPs can assume intermediate structures or random coiled but still display good anti-cancer activity such as proline-arginine-rich and tryptophan-rich peptides. Melittin, Bovine lactoferrin (LfcinB) and Alloferon are examples of alpha-helical, β-sheet and random-coiled anticancer peptides, respectively [91,92].

#### 3.2.5. Hydrophobicity, Amphipathicity and Hydrophobic Moment

The presence of a threshold percentage of hydrophobicity is necessary to perform the biological activities in ACPs [93]. Nearly all ACPs, e.g., Temporin A exhibit moderate hydrophobic moments and amphipathic conformations upon interaction with target membranes [49]. The amphipathicity in β-sheet ACPs can be created by well-organized polar and nonpolar surfaces [94]. It is observed that an increase in hydrophobicity at certain positions in the sequence of a peptide may promote peptide amphipathicity. Generally, this phenomenon results in the enhancement of mean hydrophobicity of peptides, but the site-specific hydrophobicity is proportional to its hydrophobic moment and subsequently, the amphipathic characters of peptides.

#### 3.2.6. Polar Angle

Polar angle is an estimation of the relative distribution of polar and nonpolar residues on two opposite faces of a peptide in an amphipathic helix. Increased segregation between hydrophilic and hydrophobic domains of the peptide would increase the polar angle as in VmCT1 analogs [95]. The polar angles are correlated with the net ability of the peptide to induce lethal pores in membranes, and peptides with smaller polar angles stimulate unstable pore formation than peptides with larger polar angles [96].

#### 3.2.7. Peptide Self Assembly

Self-assembling peptides (SAPs) are small peptide sequences alternating in hydrophilic and hydrophobic amino acid residues [97]. Peptide self-assembly is a process in which peptides spontaneously form ordered aggregates when they encounter different microenvironments [98]. Many physical and chemical interactions stabilize this state [99]. Self-assembling peptides such as RADA16 and E3PA (AAAAGGGEEE) have great potential in cancer therapy as they can be utilized for cancer cell targeting, forming nanostructures, drug delivery, etc. [100,101,102].

#### 3.2.8. Chemical Modifications

Modifications, both at the side chain and the main chain have been reported to improve anti-cancer activity [103,104]. The replacement of natural amino acids with non-natural amino acids in the main chain has been done by investigators to improve biological activity. Similarly, many chemical modifications in the side chain such as PEGylation, phosphorylation, adding fatty acids and glycosylation have been reported to facilitate the entry of the peptide into the target cancer cell that thereby reduces the IC50 values of the concerned peptide [105,106]. Lee et al. have found that PEGylation of a cationic antimicrobial decapeptide KSL-W (KKVVFWVKFK) improves the survival of mice in a sepsis model. It was also shown that the chemically modified peptide nullified the lipopolysaccharide (LPS)-induced inflammatory responses in human umbilical vein endothelial cells compared with unmodified KSLW [107].

### 3.3. Role of Non-Natural Amino Acids in Improving the Anti-Cancer Activity

Non-natural amino acids contain structural and biochemical properties that are unique to them and are not present in naturally occurring amino acids. The introduction of non-natural amino acids and other chemical groups impart conformational flexibility to the peptide which in turn improves their selectivity to cancer cells. The unnatural amino acids also evoke immune responses against the cancer cells. The substitutions of the natural amino acids Ala and Leu with their unnatural analogs β-alanine and nor-Leu make them unrecognizable by the proteolytic enzymes in the body which increases their half-life and bioavailability [108]. Tørfoss et al. showed that the cyclization of short peptides improves the anti-cancer activity of peptides [109]. They showed that the IC50 values of cyclic peptides reduce drastically against the Ramos cancer cells as opposed to their linear derivatives. Another simple change to design anti-cancer peptides with high therapeutic indexes is the introduction of d-amino acids. Substituting the l-amino acids with the corresponding d-amino acids results in structural alterations which make the peptide less hemolytic and cytotoxic to normal cells and also significantly improve the proteolytic stability of the peptide [110]. 1,2,3,4-tetrahydroisoquinoline-3-carboxylic acid, also known as Tic falls within the category of non-natural chiral α–amino acids (Figure 2). Tic can be used to design peptides that can be used for their dual role of Anti-cancer and immunomodulatory properties. Owing to the distinctive geometrical conformation, it can be used as a substitute for aromatic amino acids like tyrosine and phenylalanine and the imino acid proline to design conformationally constrained analogs [111]. Besides that, the percent hydrophobicity of Tic is more than that of proline hence it can be used to enhance the percent hydrophobicity of the peptide, which is an important biophysical parameter for biological activity. Octahydroindole-2-carboxylic acid (Oic) is another α-amino acid having a bicyclic structure and can be used instead of proline to impart rigidity to the peptide backbone. Oic is more lipophilic than proline and hence its introduction in the peptide design may improve the absorption and distribution through the cell membranes [112]. Amnoisobutyric acid, (Aib) is known to induce helical structures in peptides [113]. Since helicity is an important property of many anticancer and immunomodulatory peptides, it can be used at strategic positions in place of classical amino acids to design anti-cancer and immunomodulatory peptides. Alanine and leucine residues are very common in ACPs. However, to improve the activities, both amino acids can be substituted with 1-aminocyclohexane carboxylic acid (A6c)/A5c. Similarly, 2,4-diaminobutanoic acid/2,3-diaminopropionic acid can be used for lysine to design less cytotoxic analogs with improved biological activities [94,114]. It is worth mentioning that the design of amino acid analogs is as important as designing the biologically active anti-cancer peptides [115,116]. The introduction of non-natural amino acids readily improves the anti-cancer activity, which can help to speed up the translation of ACPs to clinical settings.

## 4. Cell Selectivity

The difference in the membrane composition of a cancer cell versus a normal cell plays a very crucial role to dictate the membrane binding and consequent anti-cancer activity of cationic amphiphilic peptides. In particular, membranes of different cancer cells contain phosphatidylserine, sialic acid-containing lipids and proteins, and heparan sulfate, which impart an overall anionic nature to it. Contrary to it, the non-cancerous cells are zwitterionic in nature due to the presence of phosphatidylcholine and sphingomyelin [117] (Figure 3A). The cell surface of almost all the cancer cell membranes displays phosphatidylserine. The latter is present in the inner leaflet of the lipid bilayer in normal cells in sharp contrast to the cancer cells. This has led to the development of the idea that phosphatidylserine could be used as a diagnostic cancer marker [118]. Apart from membrane composition differences, microvilli surface area is also seen to be high in cancer cells which could also be potential targets of cationic amphiphilic peptides [119,120]. Leon et al. while working on a synthetic ACP viz. HB43 also called FLAK50 (FAKLLAKLAKKLL) showed that phosphatidylserine (PS) plays an important role in the cell selectivity of the peptide in distinguishing a cancerous cell from a non-cancer cell [121]. Using various biophysical and in silico approaches the authors show that lysine side chains of HB43 and the carboxylate group of phosphatidylserine catalyze the alpha-helical conformation that facilitates its internalization in cancer cells. Membrane permeabilization assays demonstrated that the peptide-membrane interaction may lead to the destabilization of PS-containing vesicles with respect to PC-containing ones, which were used as non-cancerous membrane mimetic vesicles. What did the authors not investigate could be the immunomodulatory activity of the peptide.HB43 is a leucine-lysine rich peptide. Rosenfeld et al. and Azmi et al. have shown that such peptides can have a strong ability to inhibit the LPS-mediated inflammation by directly binding to the peptide [122,123]. Electrostatic interaction between the anionic cancer cell membrane and the anti-cancer peptides is a well-established and widely studied mechanism of APCs. This is the first interaction that occurs between the two. Koo et al. while studying the biophysical characterization of LTX-315 (K-K-W-W-K-K-W-Dip-K-NH2) Anticancer Peptide found that electrostatic interactions were the main mechanism for the peptide’s anti-cancer activity. Their results showed that the cationic LTX-315 peptide selectively disrupted negatively charged phospholipid membranes to a greater extent than zwitterionic or positively charged phospholipid membranes [124]. Camilio et al. harnessed the immunomodulatory activity of the LTX-315 peptide by using it in combination with doxorubicin [125]. LTX-315 displayed a strong additive antitumoral effect in combination with doxorubicin and induced immune-mediated changes in the tumor microenvironment. Their results displayed a complete regression of breast tumors grown from 4T1 cells in the majority of animals treated. Furthermore, imaging techniques and histological examination showed that the combination induced strong local necrosis, followed by an increase in the infiltration of CD4^+^ and CD8^+^ immune cells into the tumor parenchymal tissue. In an independent study, Sveinbjornsson et al. showed that LTX-315 induces ICD through its membranolytic mode of action, leading to the release of potent immunostimulants in addition to a wide spectrum of tumor antigens, thus creating an essential premise for tumor-specific immune responses [126]. Intratumoral treatment with LTX-315 resulted in complete regression of orthotopic B16 melanomas in 80% of animals [126].

## 5. Mechanism of Membrane Targeting and Entry to the Cell

All classes of Anti-cancer peptides interact differently with the cancer cell membrane. However, in all the models suggested, either the peptides form a pore through which the cytoplasmic content eludes out, or they can directly penetrate and enter the cell, i.e., the cell-penetrating peptides (Figure 3B). The following are four different mechanisms that facilitate the entry of ACPs.

### 5.1. Barrel-Stave Model

Ehrenstein & Lecar proposed the ‘barrel-stave’ model in 1977 [127]. ACPs reach the cancer cell membrane and accumulate as monomers on the surface, which then oligomerizes because it is energetically unfavorable for a single amphipathic α-helix or β-sheet to transverse the membrane as a monomer and after that, it forms pores followed by the formation a ring-like pattern on membrane exteriors. Later they align perpendicularly to the membrane. In their perpendicular arrangement, the peptides begin to insert into the lipid core of the cell membrane resulting in a shape of a barrel whose staves are the α-helix or β-sheet of peptides. In this model, the membrane is neither deformed nor bent during the precise drilling like the insertion process by ACPs. Alamethicin is known to respond to this model.

### 5.2. The Carpet Model

Pouny et al. proposed the carpet model in which the peptides aggregate onto the bilayer surface in a parallel manner by keeping their hydrophobic surfaces aligned towards the target cell membranes and they maintain their parallel alignment to the membrane throughout the process [128]. ACPs then intercalate to the membrane in a detergent-like manner and leading the membrane to break into small pockets and micelles. In contrast to the barrel stave and toroidal pore model, no particular pore-forming stage is evidenced in the carpet mechanism, and peptides practically never insert into membranes. Dermaseptin and LL-37 peptides are known to follow this model [129,130].

### 5.3. Toroidal Pore Model

This model has been studied in some antimicrobial peptides viz. magainin II, protegrin-1 and cecropin A [131]. A primary difference between the toroidal pore and barrel-stave models is that in the toroid pore, lipids are intercalated with peptides in the transmembrane channel. In this model, peptides in the extracellular environment take an α-helical structure as they interact with the anionic and hydrophobic cancer cell membrane. The bound peptides create a breach in the membrane and induce a continuous bend resulting in the formation of toroidal pores.

### 5.4. Cell-Penetrating Mechanism

These are the class of anti-cancer peptides that do not operate at the membrane level. Instead, they are internalized through the plasma membrane and affect the enzymatic activity and intracellular targets such as DNA and RNA [132]. In addition to being rich in histidine, lysine and arginine, the cell-penetrating peptides are commonly known to have a lipophilic and hydrophilic tail that facilitates their translocation across the membrane [133]. Many researchers believe that the cell-penetrating peptides can be used to transport cargo into the cells and can be harnessed for their target delivery. There are web servers to predict the cell-penetrating peptides [134].

## 6. Design of Anti-Cancer Peptides as Vaccines to Influence the Immune System

Due to the complex pathophysiology of cancer, the development of peptide vaccines has always been a challenge [135]. However, various in silico approaches have been made to design anti-cancer peptide vaccines. To design the anti-cancer peptide vaccines, the initial step is to identify the antigenicity of the target protein. This can be done by utilizing softwares like ANTIGENpro or VaxiJen [136,137]. It has been observed that the former is more specific and precise than VaxiJen and the resultant probability indicated high antigenicity of the protein [136]. The B-cell epitopes on protein antigens can be determined by using Kolaskar and Tongaonkar antigenicity scale [138]. T-cell epitopes can be selected by NetCTL prediction server [139]. The three-dimensional structure of the epitope can be predicted by PEP-FOLD [140]. The immunogenic peptide thus created might be used to mount an immune response against the tumor cell epitopes. (Figure 4). The use of combinatorial technologies such as using page display libraries can also be utilized to identify peptide molecules that can bind to receptors on the cancer cell surface for cell internalization [141,142,143]. The molecules selected in this way can be subsequently synthesized and modified to obtain peptide drugs with high affinity for the target molecule. In one approach, targets associated with inflammation can be used to screen out therapeutic peptides from a random phage library (usually Ph. D. -7 library) without making a phage library. Another method involves constructing the phage that displays candidate peptides on the surface. After that, an affinity selection technique termed biopanning can be used to select peptides that bind to a given target of interest.

## 7. Limitations of Anti-Cancer Peptides with Immunomodulatory Activity and Plausible Resolution

There has been promising development in the field of cancer to push anti-cancer peptides as the lead molecules in cancer therapeutics. Most of the anti-cancer peptides preferably bind to the cancer cells more than normal cells due to their cationic nature. However, the observed anti-cancer effects of the peptides depend mostly on compositional differences between cancer and non-cancer cells, where the former frequently display higher content of negatively charged lipids and other membrane components, such as phosphatidylserine and gangliosides, which result in higher peptide binding and membrane insertion, in turn destabilizing such membranes. The short plasma life of peptides is another drawback that hinders the clinical translation of anti-cancer peptides. Like cancer cell membranes, there are many anionic compounds present in blood that can suppress the anti-cancer activity of the peptides. Naturally occurring peptides like melittin, chrysophsin, etc. are equally cytotoxic to both normal and cancer cells [47,144]. In addition to this, many anticancer peptides work at higher concentration ranges and hence chemical modifications are needed to lower their activity concentration, which increases their cost of production.

Hence, more peptide analogs need to be made that could retain the anti-cancer activity and display low cytotoxicity to normal cells. An emphasis should be given to the synthesis and incorporation of non-natural amino acids to increase the plasma life of the designed peptides. More emphasis needs to be given to synthesizing peptides using conjugations. A combined effort to create more open peptide databases should also be made to know the amino acid compositions that make bioactive peptides. It will also reduce the “trial and error effort” to a large extent. This also underlines the need to perform more in vivo experiments on more complex animal models to accurately evaluate the true potential of anti-cancer peptides.

## 8. Summary and Concluding Remarks

During COVID-19 pandemic, antiviral peptides became a center of attraction due to the development of peptide vaccines targeted against SARS-CoV-2 [145,146]. Although the pandemic is mostly over, however it emphasized the need for harnessing the potential of therapeutic peptides. Diseases like cancer are always a challenge and its deep connection to inflammation provides an opportunity to harness peptides as both anti-cancer and immunomodulatory agents. A combined and collaborative approach of the latest technologies such as immunoinformatic characterization, computer-assisted drug design (CADD) and epitope-based design along with the PK/PD experiments and animal studies can help in the development of new peptide molecules. Such rationally designed peptides can target both the inflammatory and the cancer nodes of the disease and translate to clinical settings.

## Figures and Tables

**Figure 1 pharmaceutics-14-02686-f001:**
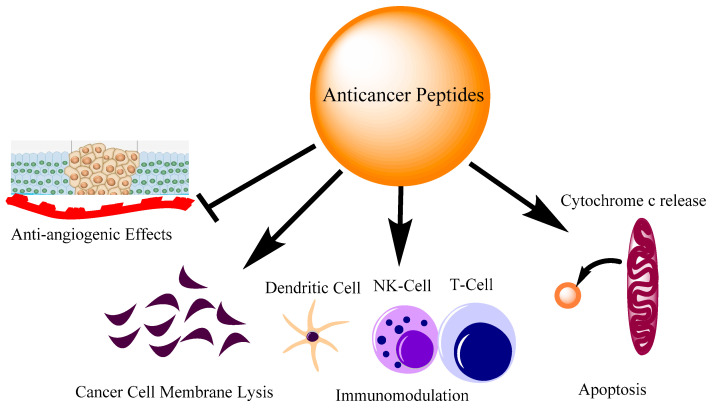
Different modes of action of anti-cancer peptides. The action of peptides involves inhibiting angiogenesis, direct cell membrane lysis, immune cell regulation and apoptosis by cytochrome c release from mitochondria.

**Figure 2 pharmaceutics-14-02686-f002:**
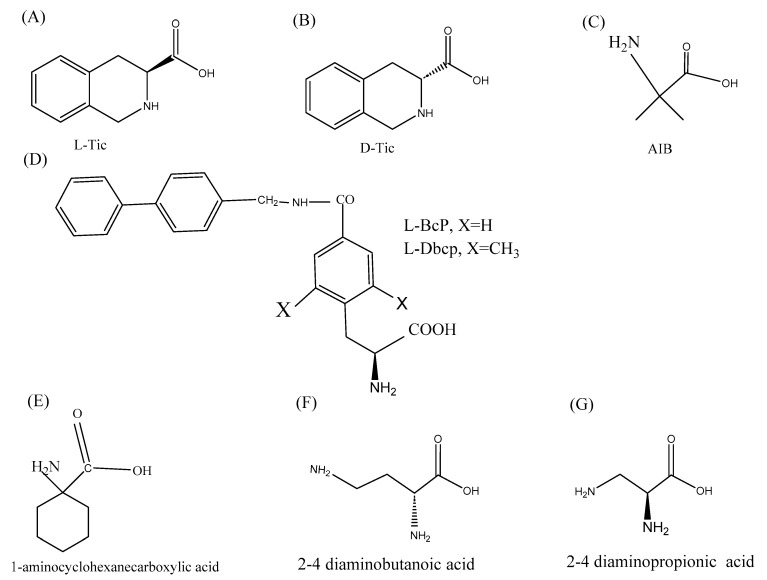
(**A**–**G**), Non-natural amino acid analogs and chemical compounds that can be used to alter the secondary structure and improve the anti-cancer activity of the peptides.

**Figure 3 pharmaceutics-14-02686-f003:**
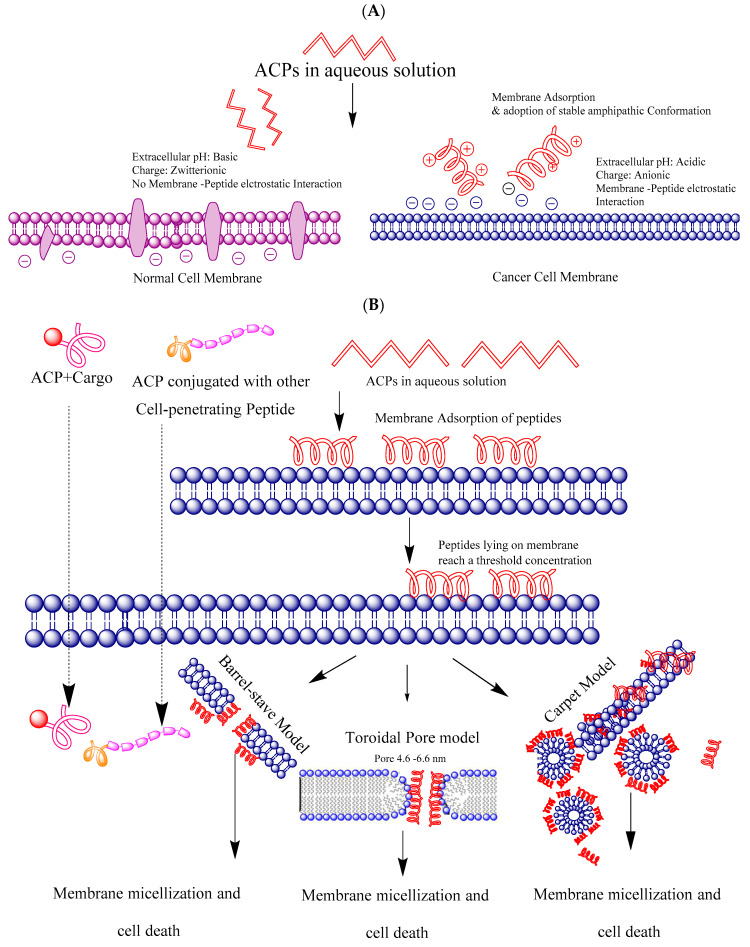
Cell Selectivity and Models of cancer cell membrane permeation by ACPs. (**A**) ACPs fail to attain the stable secondary structure in a normal cell membrane microenvironment due to zwitterionic charge and basic pH. The ACPs interact with anionic lipids of the cancer cell membrane to initiate anti-cancer activity. (**B**) The peptides directly penetrating the membrane can be used to deliver cargo into the cell. The non-permeable peptides can be conjugated with Cell-penetrating peptides (CPP) to facilitate their entry into the cell. Other peptides can breach the membrane integrity by Barrel-stove, Toroidal pore, or carpet model after interacting electrostatically with the cancer cell membrane.

**Figure 4 pharmaceutics-14-02686-f004:**
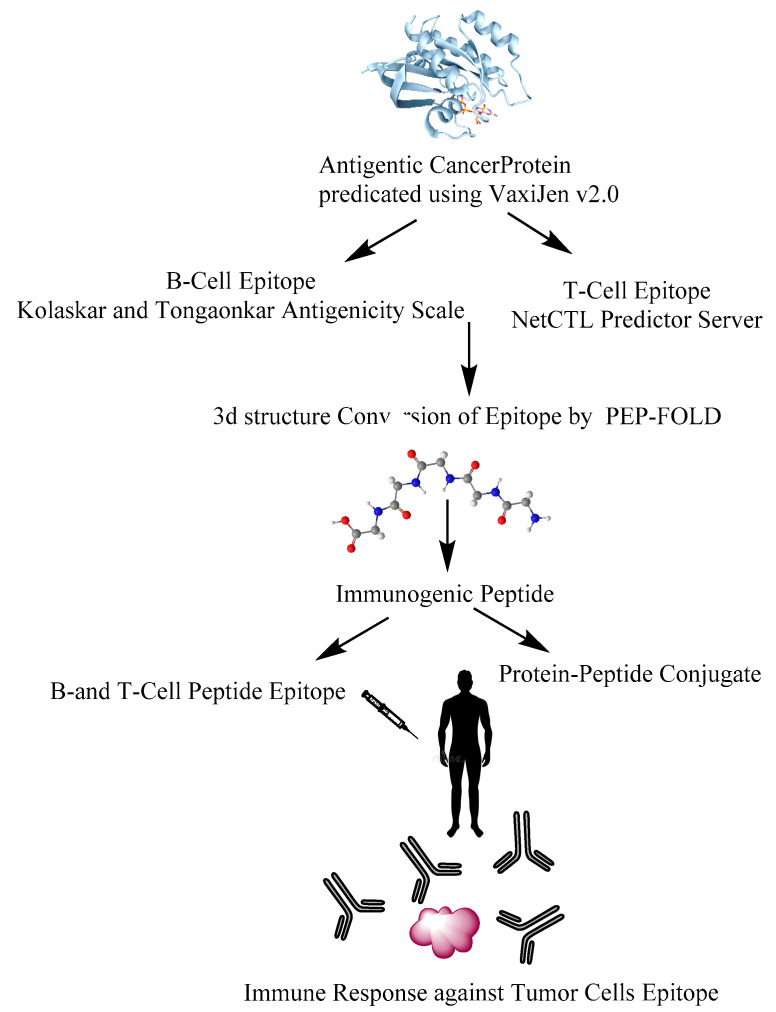
A simplified schematic diagram to design peptide vaccines against tumor cells.

**Table 1 pharmaceutics-14-02686-t001:** Peptide and the amino acid sequences discussed in the Review.

Sl No	Peptide Name	Sequence
1	LL-37	LLGDFFRKSKEKIGKEFKRIVQRIKDFLRNLVPRTES
2	Magainin II	GIGKFLHSAKKFGKAFVGEIMNS
3	Melittin	GIGAVLKVLTTGLPALISWIKRKRQQ
4	P18	KWKLFKKIPKFLHLAKKF
5	Tritrpticin	VRRFPWWWPFLRR
6	Indolicidin	ILPWKWPWWPWRR
7	PuroA	FPVTWKWWKWWKG
8	Chrysophsin-1	FFGWLIKGAIHAGKAIHGLIHRRRH
9	Chrysophsin-2	FFGWLIRGAIHAGKAIHGLIHRRRH
10	Temporin L	FVQWFSKFLGRIL
11	Temporin A	FLPLIGRVLSGIL
12	Bombinin H2	LIGPVLGLVGSALGGLLKKI
13	Hepcidin	ICIFCCGCCHRSKCGMCCKT
14	KSL-W	KKVVFWVKFK
15	HB43	FAKLLAKLAKKLL
16	LTX-315 *	KKWWKKW-DipK
16	KTH-222	LKGQLRCI
17	K4R2-Nal2-S1 **	KKKKRR-Nal-Nal-KKWRKWLAKK
18	PR-39	RRRPRPPYLPRPRPPPFFPPRLPPRIPPGFPPRFPPRFP
19	L-K6	IKKILSKIKKLLK
20	IK-13	CIIKKIIKKIIKK
21	LK-13	CLLKKLLKKLLKK
22	Alloferon	HGVSGHGQHGVHG
23	LactoferricinB (LfcinB)	FKCRRWQWRMKKLGAPSITCVRRAF
24	RADA16	RADARADARADARADA
25	E3PA	AAAAGGGEEE
26	FLAK50	FAKLLAKLAKKLL
27	VmCT1	FLGALWNVAKSVF

* Dip is β-diphenylalanine. ** Nal is β-naphthylalanine.

## Data Availability

The data supporting the findings of this study are available from the corresponding authors upon reasonable request.

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
