# Peer review of "Role of Anti-Cancer Peptides as Immunomodulatory Agents: Potential and Design Strategy"

_pharmaceutics, 2022, doi:10.3390/pharmaceutics14122686_

Round 1

Reviewer 1 Report

Title: Role of Anti-cancer Peptides as Immunomodulatory agents: Potential and Design Strategy

 In the present manuscript entitled above, authors enumerated many anti-cancer peptides, explained their mechanism of action, assessed the structural and biophysical parameters, summarized the design idea of anti-cancer peptides, and put forward Limitations of anti-cancer peptides. In my opinion, the manuscript is well written and understandable. I think it will be of interest to this journal and its readers in this field. Overall, this is an interesting review, but some additional clarifications are required:

 1) Authors assessed the biophysical parameters of anti-cancer peptides determining anticancer and immunomodulatory properties, you’d better add some examples below each one.

2) You have summarized the limitations of anti-cancer peptides. Why not enumerate some improvement measures for limitations?

3) According to the title, the focus of this review should be on the design strategy of anti-cancer peptides. This part is not detailed enough.

4) The clarity of Figure 2 is insufficient

5) There are typographical (line 300-306) and syntax errors which can be picked out by the authors on a reread of the submission. Please correct them.

Author Response

In the present manuscript entitled above, authors enumerated many anti-cancer peptides, explained their mechanism of action, assessed the structural and biophysical parameters, summarized the design idea of anti-cancer peptides, and put forward Limitations of anti-cancer peptides. In my opinion, the manuscript is well written and understandable. I think it will be of interest to this journal and its readers in this field. Overall, this is an interesting review, but some additional clarifications are required:

We would like to thank the reviewer for his/her overall positive assessment of the manuscript and opinion about the suitability of the manuscript for the pharmaceutics journal.

  • Authors assessed the biophysical parameters of anti-cancer peptides determining anticancer and immunomodulatory properties, you’d better add some examples below each one.

Thank You for this great suggestion. In the revised manuscript, we have added multiple examples of the peptides in each subsection of biophysical parameters ( Section 3.2) along with the relevant references. All the changes are marked up using the “Track Changes” function of  MS Word.

  • You have summarized the limitations of anti-cancer peptides. Why not enumerate some improvement measures for limitations?

In order to address the reviewer’s comments, we have rewritten Section 7 and also edited the subheading to:

Limitations of Anti-cancer peptides with immunomodulatory activity and plausible resolution:

 We have also discussed the possible improvements and methods that can be adopted to reduce the limitations of the Anti-cancer peptides and push them forward as the lead molecules in the field of cancer therapeutics

  • According to the title, the focus of this review should be on the design strategy of anti-cancer peptides. This part is not detailed enough.

We would like to draw the kind attention of the respected reviewer to Section 6 i.e.

“Design of Anti-cancer peptides as vaccines to influence the immune system”

Here we have discussed how a wide range of bioinformatic tools can be used to design potential peptide drugs. In the revised version of the manuscript, we have added another paragraph in the same section discussing the identification of bioactive peptide molecules against a specific target using combinatorial technologies such as page display libraries. Besides this, in Section 3.1, we thoroughly discussed the biophysical and structural determinants that can be employed to design anti-cancer peptides with immunomodulatory activity. We have also added examples of various peptides that possess those structural determinants.

  • The clarity of Figure 2 is insufficient

We have fully redrawn the whole diagram to enhance the picture quality of Figure 2 in the revised version of the manuscript.

5) There are typographical (line 300-306) and syntax errors which can be picked out by the authors on a reread of the submission. Please correct them.

Thank You. It has been corrected now in the revised version of the manuscript.

Reviewer 2 Report

I revised the manuscript entitled "Role of Anti-cancer Peptides as Immunomodulatory agents: Potential and Design Strategy" by Tripathi A.K. and Vishwanatha J.K. which provides an overview of natural antimicrobial peptides with anticancer activity and synthetic chemically modified peptides with dual role (anti-cancer and immunomodulatory function). Nowadays, peptides represent a potential therapeutic  tool for several clinical applications.  In addition, the knowledge and study of their chemical properties and binding abilities to a specific target molecule by using bioinformatic tools help to design and predict the better aminoacidic composition, peptide secondary structure and interaction modes. Accordingly, candidate peptides can be used for the development of new peptide therapeutics able to act on the interconnection between tumor mass and inflammatory-immune response.

The title matches well with the content and the abstract is concise and clear. The manuscript is well written,  articulated and provides knowledges to this research field.

The work is suitable for publication in the “Pharmaceutics”  journal after minor revision.

I suggest the authors make the following changes:

1.       Line 34. Please add appropriate reference for sentence “Chemotherapy is the most established method of treatment that kill fast-dividing cancer cells. However, most cancer drugs have very poor cell selectivity and kill normal cells as well along with cancer  cells indiscriminately” .

2.       Please move reference 2 about Radio- and Immunotherapy at the end of related sentence on line 42.

3.       Figures must appear immediately after mentioning them in the text. Please correct them.

4.       Line 98. The authors refer to a table that was not provided for the review process. Please add it or if not available remove the wording “Table S1, Supplementary Information” from the text.

5.       Line 149-150 Please eliminate Bold font.

6.       Line 262-264 Please eliminate Italic font.

7.       Please add specific references in paragraph 5.1, 5.2, 5.3.

8.       Paragraph 5. Among the different mechanisms of targeting the cell membrane, tumor-targeting peptides with or without anti-inflammatory, which bind to receptors on the cancer cell surface for cell internalization have not been discussed. Peptide can be also selected using phage display libraries and subsequently synthetized and modified to obtain peptide drug with high affinity for target molecule. Please add this topic with related references:

·         Zhang K, Tang Y, Chen Q, Liu Y. The Screening of Therapeutic Peptides for Anti-Inflammation through Phage Display Technology. Int J Mol Sci. 2022 Aug 2;23(15):8554. doi: 10.3390/ijms23158554. PMID: 35955688; PMCID: PMC9368796.

·         Aloisio A, Nisticò N, Mimmi S, Maisano D, Vecchio E, Fiume G, Iaccino E, Quinto I. Phage-Displayed Peptides for Targeting Tyrosine Kinase Membrane Receptors in Cancer Therapy. Viruses. 2021 Apr 9;13(4):649. doi: 10.3390/v13040649.

·           Karami Fath, M., Babakhaniyan, K., Zokaei, M. et al. Anti-cancer peptide-based therapeutic strategies in solid tumors. Cell Mol Biol Lett 27, 33 (2022). https://doi.org/10.1186/s11658-022-00332-w.

Author Response

I revised the manuscript entitled "Role of Anti-cancer Peptides as Immunomodulatory agents: Potential and Design Strategy" by Tripathi A.K. and Vishwanatha J.K. which provides an overview of natural antimicrobial peptides with anticancer activity and synthetic chemically modified peptides with dual role (anti-cancer and immunomodulatory function). Nowadays, peptides represent a potential therapeutic  tool for several clinical applications.  In addition, the knowledge and study of their chemical properties and binding abilities to a specific target molecule by using bioinformatic tools help to design and predict the better aminoacidic composition, peptide secondary structure and interaction modes. Accordingly, candidate peptides can be used for the development of new peptide therapeutics able to act on the interconnection between tumor mass and inflammatory-immune response.

The title matches well with the content and the abstract is concise and clear. The manuscript is well-written,  articulated and provides knowledges to this research field.

The work is suitable for publication in the “Pharmaceutics”  journal after minor revision.

We express our agreement with the reviewer that bioinformatic tools play a massive role in predicting the target protein-peptide interactions, their own secondary structures, and replacing the best amino acid compositions for better biological activities. We do appreciate the fact that the reviewer found the manuscript befitting for publication in the pharmaceutics journal after minor revision.

I suggest the authors make the following changes:

  1. Line 34. Please add appropriate reference for sentence “Chemotherapy is the most established method of treatment that kill fast-dividing cancer cells. However, most cancer drugs have very poor cell selectivity and kill normal cells as well along with cancer  cells indiscriminately” .

We have added two relevant references in support of the above statement. They are references 2 and 3 in the revised version of the manuscript.

References:

2          Mitchison, T.J. The proliferation rate paradox in antimitotic chemotherapy. Mol Biol Cell 2012, 23, 1-6, doi:10.1091/mbc.E10-04-0335.

3        Liu, B.; Ezeogu, L.; Zellmer, L.; Yu, B.; Xu, N.; Joshua Liao, D. Protecting the normal in order to better kill the cancer. Cancer Med 2015, 4, 1394-1403, doi:10.1002/cam4.488.

  1. Please move reference 2 about Radio- and Immunotherapy at the end of related sentence on line 42.

Corrected! The mentioned reference is in line 43 of the revised manuscript as suggested by the reviewer.

  1. Figures must appear immediately after mentioning them in the text. Please correct them.

We have moved figure 1 just underneath the introduction which is it’s the right place.  Figure 2, which shows the structure of non-natural amino acids is placed underneath the paragraph discussing the role of non-natural amino acids. Figure 3, mentioning the Cell-Selectivity and Models of cancer cell membrane permeation by ACPs is placed underneath sections 4 and 5 which discuss cell-selectivity and various modes of action of ACPs.

  1. Line 98. The authors refer to a table that was not provided for the review process. Please add it or if not available remove the wording “Table S1, Supplementary Information” from the text.

In the first submission, we added the table providing important information about all the peptides discussed in the manuscript. It’s unfortunate that due to technical limitations, the reviewer did not have access to that table. In the revised version, we have added the table in the main manuscript body for your kind perusal. It is labeled as Table 1 in the main manuscript now in Line 524 of the revised version.

  1. Line 149-150 Please eliminate Bold font.

Thank You. We have corrected it as suggested by the reviewer.

  1. Line 262-264 Please eliminate Italic font.

Thank You. We have corrected it as suggested by the reviewer.

  1. Please add specific references in paragraph 5.1, 5.2, 5.3.

 Thank You for the great suggestion. We have added specific references in all the models proposed for peptides to work. We have also added the name of peptide candidates that are known to follow the specific model.

  1. Paragraph 5. Among the different mechanisms of targeting the cell membrane, tumor-targeting peptides with or without anti-inflammatory, which bind to receptors on the cancer cell surface for cell internalization have not been discussed. Peptide can be also selected using phage display libraries and subsequently synthetized and modified to obtain peptide drug with high affinity for target molecule. Please add this topic with related references:

  • Zhang K, Tang Y, Chen Q, Liu Y. The Screening of Therapeutic Peptides for Anti-Inflammation through Phage Display Technology. Int J Mol Sci. 2022 Aug 2;23(15):8554. doi: 10.3390/ijms23158554. PMID: 35955688; PMCID: PMC9368796.
  • Aloisio A, Nisticò N, Mimmi S, Maisano D, Vecchio E, Fiume G, Iaccino E, Quinto I. Phage-Displayed Peptides for Targeting Tyrosine Kinase Membrane Receptors in Cancer Therapy. Viruses. 2021 Apr 9;13(4):649. doi: 10.3390/v13040649.
  • Karami Fath, M., Babakhaniyan, K., Zokaei, M. et al. Anti-cancer peptide-based therapeutic strategies in solid tumors. Cell Mol Biol Lett 27, 33 (2022). https://doi.org/10.1186/s11658-022-00332-w.

We would like to thank the reviewer for pointing out this important method of identifying peptide molecules. We have added a paragraph mentioning the suggestions from lines 484-494. We have also cited all the papers mentioned by the reviewer. All the changes are marked up using the “Track Changes” function of  MS Word.

Round 2

Reviewer 1 Report

The questions I have pointed out have been carefully revised or answered and I now agree to publish the paper.